# Self-Supervised and Unsupervised Multispectral Anomaly Detection for Unknown Substance and Surface Defect Identification

Cansu Beyaz*[1,2], Mohamed Farag[2], Peer Schütt[3], Tobias Hecking[3], Jonas Grzesiak[4], Christoph Geiß[4], and Ribana Roscher[2]

[1]Fraunhofer Institute for Machine Tools and Forming Technology IWU
[2]Remote Sensing Group, Institute of Geodesy and Geoinformation, University of Bonn
[3]Institute of Software Technology, German Aerospace Center (DLR)
[4]Institute of Technical Physics, German Aerospace Center (DLR)

## Abstract

Autonomous systems and environmental monitoring require reliable detection of unknown hazardous materials to prevent traffic accidents and ecological damage resulting from chemical spills, fuel leaks, and agricultural runoff. Traditional detection methods, such as gas chromatography, pose contamination risks and cause delays, while laser-based techniques rely on prior localization of potential hotspots. This paper addresses the automatic detection of unknown materials (e.g., fertilizer, sand, soil) and surface anomalies (e.g., cracks, holes) without requiring labeled anomaly examples during training. We employ unsupervised and self-supervised deep learning methods to learn normal patterns and identify deviations. Specifically, we evaluate four models: convolutional and vision transformer-based (ViT) autoencoders, and two self-supervised methods, SimCLR and Barlow Twins. Experiments conducted on multispectral road images from the German Aerospace Center and the MVTec hazelnut dataset demonstrate that the ViT-based autoencoder outperforms its convolutional counterpart, while Barlow Twins achieves superior anomaly localization compared to SimCLR. These results indicate that reconstruction-based ViTs and redundancy-reducing self-supervision are promising strategies for anomaly detection in road safety and environmental protection.

## 1 Introduction

The World Health Organization (WHO) reported in 2016 [1] that 13.7 million deaths (24% of global deaths) and 23% of the global disease burden were linked to modifiable environmental factors such as chemicals, waste, and pollution. Exposure to selected chemicals alone accounted for an estimated 1.6 million deaths, although evidence on specific chemical risks is still emerging.

In Europe, 342,000 contaminated sites were identified in 2014 (5.7 per 10,000 inhabitants), with waste disposal (municipal and industrial) being the main source of soil and groundwater contamination [2]. In Africa, the WHO estimates that one-third of the disease burden is attributable to environmental risk factors, with hazardous waste ranking among the top three concerns. Accordingly, the detection of hazardous materials is not only a technical challenge but also a critical public safety and environmental health priority.

Traditional approaches include visual inspection, chemical sensors, and basic computer vision techniques, but are limited by high costs, subjectivity, and restricted detection capabilities across different spectral ranges. Recent deep learning-based anomaly detection methods [3] hold promise for reducing reliance on manual inspection. However, detecting unknown materials and surface anomalies without labeled anomalies remains challenging, since existing approaches often rely on expensive inspection and assumptions with poor generalization.

Recently, Schütt et al. [4] proposed an unsupervised approach leveraging a convolutional autoencoder, demonstrating promising results. We extend this line of research by investigating both unsupervised and self-supervised anomaly detection, testing contrastive methods on RGB data to enable future evaluation on multispectral data.

> **Main Hypothesis**
>
> Unsupervised and self-supervised learning methods can effectively learn normal surface patterns and therefore detect anomalies and unknown hazardous materials without requiring labeled anomaly examples.

The proposed framework evaluated four distinct deep learning approaches for anomaly detection, as illustrated in Figure 1. Unsupervised methods utilize autoencoder architectures with ResNet [5] and Vision Transformer (ViT) [6] encoders, while two

*Corresponding Author. Email: cansu.beyaz@iwu.fraunhofer.de

Proceedings of the 7th Northern Lights Deep Learning Conference (NLDL), PMLR 307, 2026.

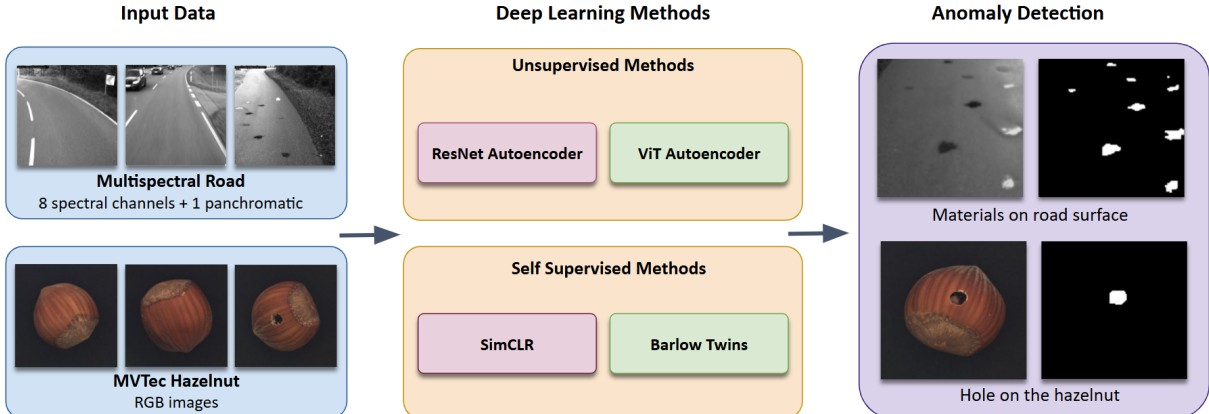

**Figure 1. An overview of the proposed anomaly detection framework**, comparing four deep learning methods across two evaluation datasets.

self-supervised approaches implement SimCLR [7] and Barlow Twins [8] techniques. The framework is designed to handle diverse input modalities and generate binary anomaly maps that localize and segment anomalous regions. We first benchmark the four approaches on the well-established MVTec AD Hazelnut (RGB) dataset [9]. We select the best-performing approach and evaluate it on the DLR multispectral road dataset (8 VIS/NIR spectral bands + 1 panchromatic) to detect unknown surface materials (e.g., fertilizer, sand, soil).[1]

In summary, our contributions are threefold:

---

**Main Contributions**

**(i)** To the best of our knowledge, we present the first comparision of Vision Transformer-based and CNN-based autoencoders with contrastive self-supervised learning methods (SimCLR and Barlow Twins) for surface defect detection; **(ii)** we show that ViT-based autoencoder outperforms a ResNet-based autoencoder and a convolutional autoencoder in this task; and **(iii)** we demonstrate that Barlow Twins surpasses SimCLR for anomaly localization, showing particular promise in computationally constrained settings or when training data is limited.

---

Supported by experimental results on both multispectral road images and the MVTec hazelnut dataset. It underlines the potential of such detection methods to prevent accidents, reduce exposure to toxic substances, and mitigate long-term contamination risks.

The code can be provided upon request.

# 2   Related Work

**Anomaly detection (AD) in Computer Vision.** It is a subtask of the generalized Out-of-Distribution (OoD) detection problem [10], aiming to identify unusual patterns that deviate from normal data at test time. Such deviations may result from *covariate* or *semantic* shifts.[2] Unlike OoD detection, AD does not require distinguishing between different in-distribution (ID) classes, treating them as a single group. AD has broad applications, including adversarial defence and industrial inspection.

**Anomaly Detection Approaches.** Multiple methods have been proposed for anomaly detection [10], among which we focus on *reconstruction*-based and *distance*-based approaches. In reconstruction-based methods, an encoder–decoder architecture is trained on in-distribution (ID) samples to reconstruct them accurately; deviations in reconstruction error indicate potential anomalies. In distance-based methods, anomalous samples are expected to lie far from the centroids of ID clusters in the feature space. By thresholding a distance metric, such as Mahalanobis or Euclidean distance, anomalies can be identified.

Autoencoders (AEs) [3] are widely used in reconstruction-based approaches, compressing inputs into a low-dimensional latent space and then reconstructing them from this representation. For distance-based methods, Hojjati et al. [12] provide a comprehensive overview of the role of self-supervision in anomaly detection. One important family is contrastive learning, where the model is trained to bring similar samples closer and push dissimilar ones apart, thus regularizing the embedding space to prevent anomalous embeddings from collapsing onto ID embeddings. This principle, referred to by Postels et al. [13] as *informative representa-*

---

[1]VIS: visible spectrum; NIR: near-infrared spectrum.

[2]In this paper, we focus on semantic shift, defined by Ruff et al. [11] as images containing objects from non-normal classes.

*tion* regularization, enhances separability between ID and anomalous data.

**Multispectral Imaging.** It captures information across spectral bands beyond the visible range [14]. Different materials exhibit unique spectral signatures that are often invisible in standard RGB images, making multispectral imaging valuable for material identification and anomaly detection [15]. Chen et al. [16] demonstrated this potential by combining near-infrared hyperspectral imaging with convolutional neural networks for standoff material identification. In agriculture, Strothmann et al. [17] used convolutional autoencoders to detect anomalous grapevine berries from multispectral data. More recently, Wang et al. [18] introduced attention mechanisms for multispectral anomaly detection, enabling models to focus on the most discriminative spectral bands for each task. Schütt et al. [4] demonstrate that combining convolutional Autoencoders with multispectral imaging enhances anomaly detection performance; specifically, they show that using NIR as input outperforms models relying solely on the RGB spectrum. This finding motivates our own experiments in a similar direction.

**Hazardous Material Detection.** Existing methods for detecting hazardous materials often rely on RGB or multispectral data [19–22], framing the task as object detection—either targeting the materials themselves or their hazard symbols. A key limitation of these approaches is their dependence on labeled datasets and the *closed-world* assumption, where no distributional or semantic shifts are expected.

# 3 Methodology

## 3.1 Multispectral Data Capture

Data collection employs a vehicle-mounted sensor array system that is equipped with two CMS series multispectral cameras from SILIOS Technologies [23]. The cameras capture spectral ranges: visible light (VIS, 430-700 nm) and near-infrared (NIR, 650-930 nm). Each camera utilizes CMOS CS-mount technology with 5.3 $\mu$m pixel pitch, operating at up to 60 fps with 10-bit ADC precision.

The complete sensor configuration is illustrated in Figure 2, which shows the integrated vehicle-mounted system comprising the two multispectral cameras (VIS and NIR), laser-based UAV classification system (LUCS [24]), radar sensors, alignment laser, Global Positioning System (GPS) module, and brightness sensor. This comprehensive setup enables the capture of multispectral imagery alongside environmental and positioning data for anomaly detection applications.

The camera's array-type optical interface organizes pixels into 3×3 macropixels, each containing

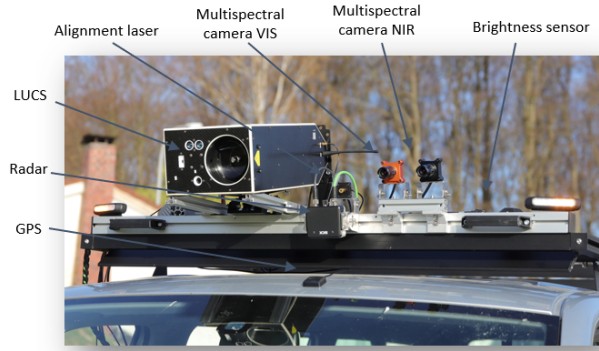

**Figure 2. Vehicle-mounted sensor array system**, showing the complete setup including LUCS, radar, GPS, alignment laser, brightness sensor, and two multispectral cameras (VIS and NIR) used for data collection.

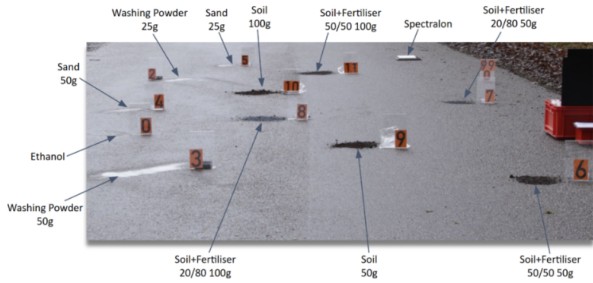

**Figure 3. Examples of substances**, applied to the road surface for anomaly detection.

eight distinct spectral filters (VIS or NIR) plus one panchromatic channel. We developed a controlled experimental protocol using visually and spectroscopically similar but environmentally safe placeholder substances. These serve as proxies for hazardous substances in our anomaly detection framework.

Substances are strategically applied to road surfaces as shown in Figure 3. These test substances include washing powder, sand, soil, fertilizer mixtures, and ethanol, each placed in controlled quantities ranging from 25g to 100g.

## 3.2 Anomaly Detection Approaches

**Unsupervised Learning with Autoencoders**. We compare two autoencoder variants for multispectral anomaly and material detection: one with a ResNet encoder [5] and the other with a Vision Transformer (ViT) encoder [6], both using a shared convolutional decoder. These models are trained exclusively on normal samples to learn a compact representation of normal appearance. During inference, anomalies are detected based on reconstruction error—higher errors indicate unfamiliar or out-of-distribution patterns.

The architectures in Figure 4, both encoder variants use a shared symmetric convolutional decoder [25]. The decoder upsamples the latent representation using transpose convolutions with $2 \times 2$

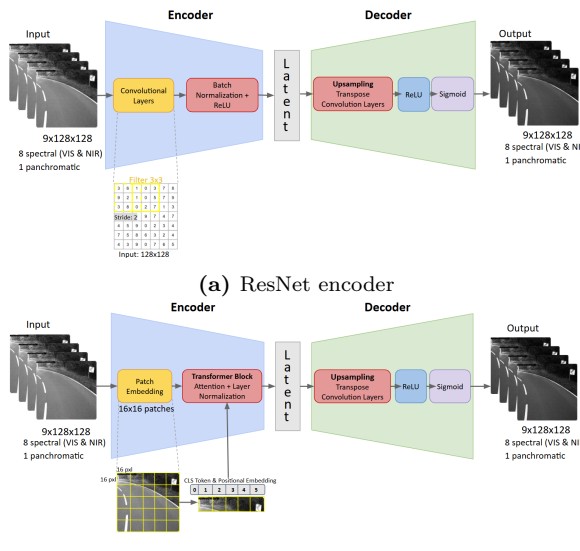

**(a)** ResNet encoder

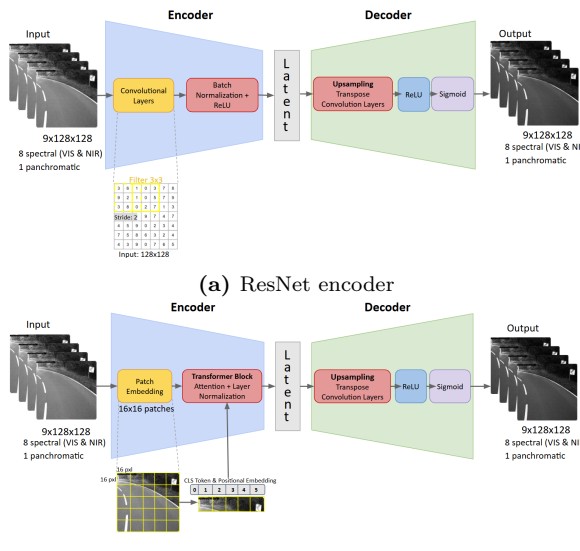

**(b)** ViT encoder

**Figure 4. Autoencoder architectures for anomaly detection**. Both encoders compress input multispectral road images ($9{\times}128{\times}128$) into latent representations. (a) ResNet uses convolutional layers with batch normalization and ReLU activation, showing $3{\times}3$ filter operation with a stride of 2. (b) ViT divides images into $16{\times}16$ patches processed by transformer blocks with multi-head attention and MLP layers, including CLS token and positional embedding.

kernels and a stride of 2. Each upsampling stage is followed by two $3 \times 3$ convolutions with batch normalization and ReLU activation. The final layer uses sigmoid activation to reconstruct the image and squash the values between 0 and 1.

**Self-Supervised Learning Methods**. Sim-CLR [7] and Barlow Twins [8] are compared for distance-based anomaly detection in this paper. For the SimCLR method, the NT-Xent (Normalized Temperature-scaled Cross Entropy) loss [7] is used:

$$\mathcal{L}_{\text{Sim}} = -\frac{1}{N}\sum_{i=1}^{N}\log\frac{\exp(\text{sim}(\boldsymbol{z}_i, \boldsymbol{z}_j^+)/\tau)}{\sum_{k=1}^{2N}\mathbf{1}_{k\neq i}\exp(\text{sim}(\boldsymbol{z}_i, \boldsymbol{z}_k)/\tau)}, \tag{1}$$

where $\boldsymbol{z}_i$ is the anchor representation, $\boldsymbol{z}_j^+$ is the positive pair (augmented view of the same image), $\text{sim}(\cdot, \cdot)$ is the cosine similarity function, $\tau$ is the temperature parameter, and $N$ is the batch size. This loss pulls positive pairs closer together while pushing negative pairs apart in the feature space.

For the Barlow Twins method, the loss function combines invariance and redundancy reduction terms [8]:

$$\mathcal{L}_{\text{BT}} = \sum_i (1 - \mathcal{C}_{ii})^2 + \lambda \sum_i \sum_{j\neq i} \mathcal{C}_{ij}^2, \tag{2}$$

where $\mathcal{C}$ is the cross-correlation matrix between the normalized representations $\boldsymbol{z}_A$ and $\boldsymbol{z}_B$ of two augmented views:

$$\mathcal{C}_{ij} = \frac{\sum_b \boldsymbol{z}_{b,i}^A \boldsymbol{z}_{b,j}^B}{\sqrt{\sum_b (\boldsymbol{z}_{b,i}^A)^2}\sqrt{\sum_b (\boldsymbol{z}_{b,j}^B)^2}}\,. \tag{3}$$

The first term encourages the diagonal elements to be close to 1 (invariance), while the second term with parameter $\lambda$ pushes the off-diagonal elements toward 0 (redundancy reduction). Barlow Twins eliminates the need for negative pairs by reducing redundancy between embedding components through cross-correlation matrix optimization.

Both methods share a common architectural foundation while differing in their learning objectives and projection strategies. The pipeline begins with normal images and applies data augmentation to create two correlated views of the same input. These augmented views are then processed through a shared ResNet50 [5] encoder for feature extraction, followed by transformation through projection heads. Figure 5 illustrates this common framework that underlies both approaches.

The learned representations from both methods serve as the foundation for anomaly detection during inference. According to Lee et al. [26], test images are processed through the trained encoder to extract features in a learned feature space that is assumed to be Gaussian. These features are then compared against this assumed normal distribution using statistical distance measures such as Mahalanobis distance to compute anomaly scores [3].

## 3.3 Anomaly Scoring and Prediction

**Anomaly Scoring**. In the reconstruction-based detection approach, the Mean Squared Error (MSE) is computed per pixel between the input image and the reconstructed output.

A multispectral image is also represented as a single combined image (as seen in the Figure 7). Reconstruction errors are calculated for each pixel across all nine spectral channels. These errors are then summed and normalized by dividing by 9, corresponding to the total number of channels—eight visible and near-infrared (VIS & NIR) bands and one panchromatic channel. This results in an averaged reconstruction error map, where each pixel value reflects the mean reconstruction error across all spectral channels.

For distance-based detection, anomaly scoring is performed using the encoder features (before the projection head). For a given test sample, the encoder produces a feature vector $\boldsymbol{f} \in \mathbb{R}^{2048}$ from the ResNet-50 backbone. We assume that the features are distributed according to a multivariate Gaussian distribution, and we fit the model to the training set features. Given the set of normal training features

---

[3]More details are available in Appendix B

$F = \{\boldsymbol{f}_1, \boldsymbol{f}_2, \ldots, \boldsymbol{f}_N\}$, the distribution parameters are estimated as:

$$\boldsymbol{\mu}^* = \frac{1}{N} \sum_{i=1}^{N} \boldsymbol{f}_i \,, \tag{4}$$

$$\mathcal{S}^* = \frac{1}{N-1} \sum_{i=1}^{N} (\boldsymbol{f}_i - \boldsymbol{\mu}^*)(\boldsymbol{f}_i - \boldsymbol{\mu}^*)^T \,, \tag{5}$$

where $\boldsymbol{\mu}^*$ represents the estimated mean vector and $\mathcal{S}^*$ represents the estimated covariance matrix of the normal feature distribution.

The anomaly score [4] for a test feature $\boldsymbol{f}_{\text{test}}$ is computed using the Mahalanobis distance [27]:

$$\mathcal{D}_M(\boldsymbol{f}_{\text{test}}) = (\boldsymbol{f}_{\text{test}} - \boldsymbol{\mu}^*)^T (\boldsymbol{S}^*)^{-1} (\boldsymbol{f}_{\text{test}} - \boldsymbol{\mu}^*) \,. \tag{6}$$

**Anomaly Prediction**. Thresholds ($T$) for each of the four deep learning methods are selected using statistical methods with validation dataset statistics:

$$T = \mu_{val} + k \cdot \sigma_{val} \,, \tag{7}$$

where $\mu_{val}, \sigma_{val} \in \mathbb{R}$ represent the mean and standard deviation of the models' pixel-wise error on the validation dataset, and $k$ adjusts sensitivity.

After the threshold calculation, we classify pixels in the test images. For each pixel at position $(x, y)$, the value $\mathcal{V}(x, y)$ is compared with the threshold $T$. $\mathcal{V}(x, y)$ represents either the reconstruction error $\mathcal{R}(x, y)$ for autoencoders or the anomaly score $\mathcal{S}(x, y)$ for self-supervised methods. If the value is higher than the threshold, the pixel is marked as anomalous. Otherwise, it is marked as normal. The rule is defined as follows:

$$\mathcal{A}(x, y) = \begin{cases} 1 & \text{if } \mathcal{V}(x, y) > T \quad (\text{anomaly}) \\ 0 & \text{if } \mathcal{V}(x, y) \leq T \quad (\text{normal}). \end{cases} \tag{8}$$

This process results in a binary anomaly map. In this map, white pixels ($\mathcal{A}(x, y) = 1$) indicate anomalous areas, and black pixels ($\mathcal{A}(x, y) = 0$) indicate normal areas.

# 4 Experiments and Results

## 4.1 Dataset

The full multispectral dataset consists of 9,552 training images (both VIS and NIR) captured on normal road surfaces. Due to hardware limitations, a subset dataset, containing 3,242 training images, is used. For the test set, we applied the placeholder substances (compare Sec. 3.1) to road surfaces and

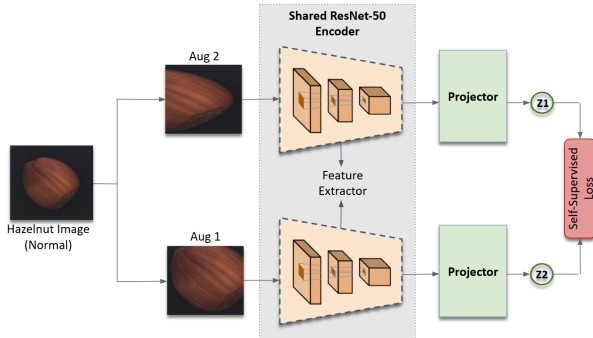

**Figure 5. SimCLR and Barlow Twins architectures for anomaly detection**. A normal hazelnut image is augmented in two different ways (Aug 1 and Aug 2) to create two correlated views. Both augmented images are processed through a shared ResNet encoder for feature extraction. A projector component transforms these features into representations $\boldsymbol{z}_1$ and $\boldsymbol{z}_2$ that are optimized according to the respective self-supervised loss functions.

captured these road areas. These images were then manually labeled using the LabelMe [28] tool. This resulted in 18 labeled test images (9 VIS, 9 NIR) for quantitative evaluation of reconstruction quality.

MVTec Anomaly Detection dataset's [9] hazelnut category for comparison, containing 391 normal training images and 70 anomalous test images with complete ground truth annotations, providing a computationally efficient benchmark for anomaly detection performance evaluation. The MVTec images are RGB images in contrast to the multispectral character of the road data.

## 4.2 Data Pre/Post-Processing

For unsupervised autoencoder-based models, MVTec images are resized to 128×128 pixels. For the multispectral road dataset, the original 1280×1024 raw images are converted into 426×339 pixel images for the 9 channels. A custom cropping function is then applied that retains the lower 60% of image height and the central 80% of width. This cropping function helps to remove irrelevant background elements, like vegetation, sky, and cars, and puts focus on the road surfaces in front of the vehicle. The multispectral images are then further resized to 128×128 to reduce training and computational time. Data augmentations are applied, including ±15° rotations, horizontal flips, and color jittering with brightness and contrast adjustments, normalized using mean=0.5, std=0.5 per channel for both datasets.

Self-supervised methods are applied to both the multispectral and MVTec datasets using 224×224 pixel inputs to match the requirements of ResNet50 [5]. Augmented views are generated through random cropping, horizontal rotation, Gaus-

---

[4] *Pixel-wise* scoring. We extract a 2048-dimensional feature for every pixel from the ResNet-50 encoder and compute its Mahalanobis distance to the training features.

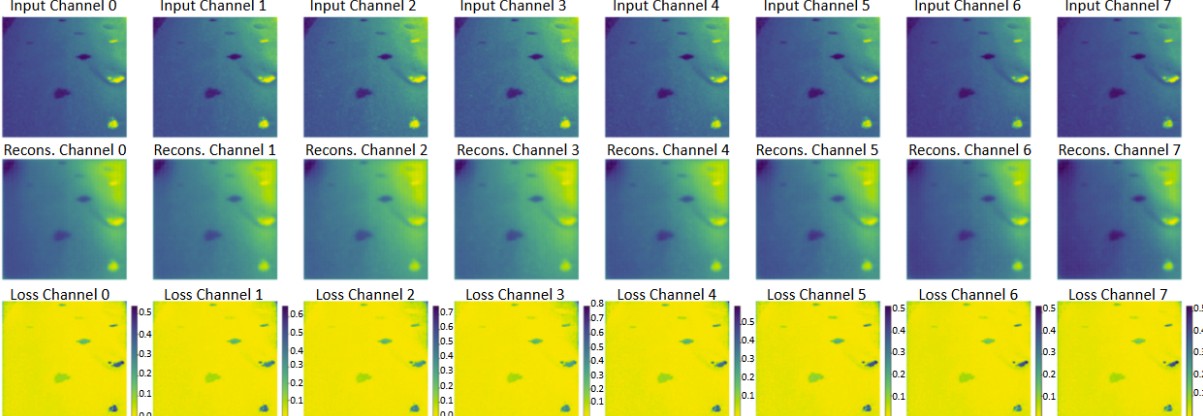

**(a) ViT-Based Reconstruction**. Qualitative Comparison Across All 8 Spectral Channels (VIS Test Image)

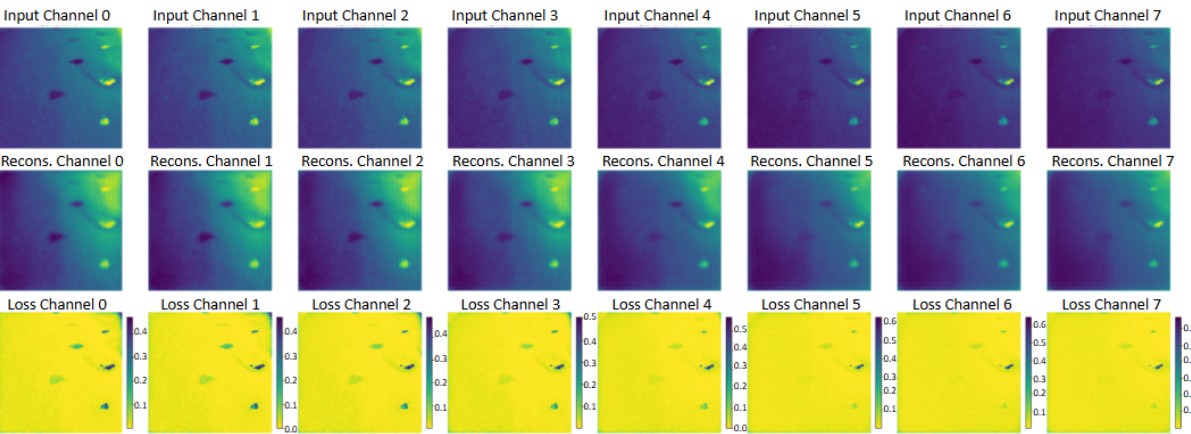

**(b) ViT-Based Reconstruction**. Qualitative Comparison Across All 8 Spectral Channels (NIR test image)

**Figure 6.** (a) VIS spectral range showing clear substance visibility across channels with varying contrast. (b) NIR spectral range demonstrating different spectral responses, with reduced contrast in higher-numbered channels.

sian blur, and color jittering, followed by normalization using mean=[0.5, 0.5, 0.5] and std=[0.5, 0.5, 0.5]. For multispectral data, the same augmentations are applied except for color jittering, which is omitted.

All image resizing operations use bilinear interpolation to maintain image quality during scale transformations. Training sets are split 80/20 for training/validation across all experiments.

For post-processing, we employ a sequence of morphological operations (closing followed by opening) using a 5×5 square structuring element to improve binary anomaly map representation. According to Martin et al. [29], closing connects separated detection pixels into meaningful regions, while opening removes noise and false positives, creating more realistic anomaly shapes that better match ground truths.

## 4.3 Evaluation Metrics

All metrics derive from the confusion-matrix outcomes (TP, FP, TN, FN) [30]. We report precision (fraction of flagged pixels that are truly anomalous), recall (fraction of all anomalies detected), and F1 (harmonic mean of precision and recall). For spatial accuracy, IoU measures overlap between predicted and ground-truth masks. To assess threshold behavior, we plot ROC curves (true-positive rate vs. false-positive rate) and summarize with AUC-ROC (0.5 = random, 1.0 = perfect). Because anomalies are rare, PR curves and AUC-PR are more informative under class imbalance [31]. Together, these metrics enable fair, comprehensive comparison of anomaly-detection methods [32].

## 4.4 Reconstructed Images

Figure 6 shows the channel-specific reconstruction quality for multispectral test images with applied substances across both VIS and NIR spectral ranges. Since VIS and NIR images are not captured simultaneously, we trained separate ViT autoencoder models for each spectral range. The figure presents reconstruction results from both the VIS-trained model applied to a VIS test image (Figure 6a) and

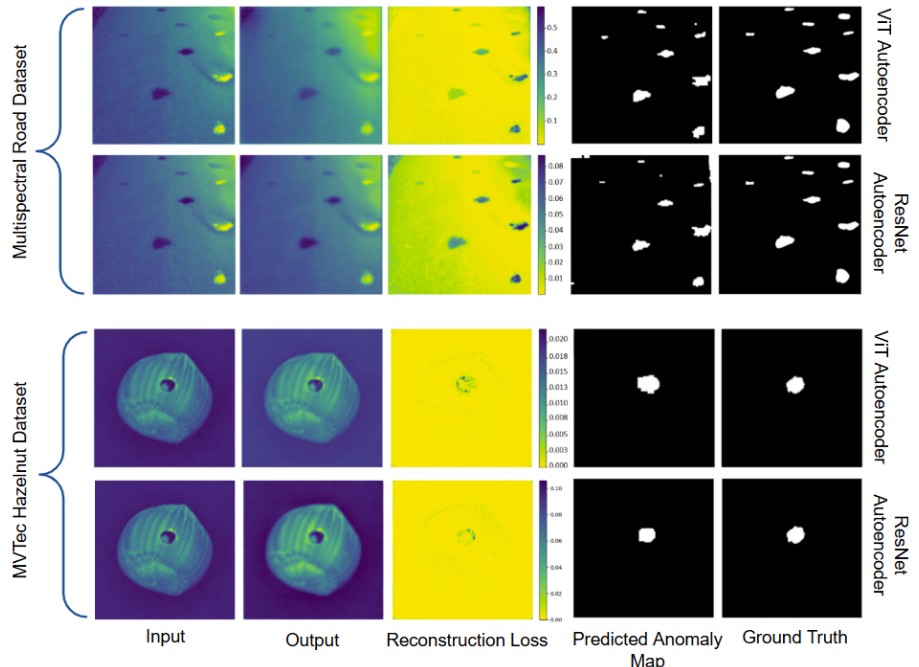

**Figure 7. Qualitative comparison of autoencoder performance**. Multispectral road dataset (top) and MVTec hazelnut dataset (bottom), the multispectral road images show various substances applied to road surfaces, while the hazelnut images demonstrate hole defect detection. Higher reconstruction loss values (darker regions) correspond to detected anomalies, which are then converted to binary masks for evaluation against ground truth.

the NIR-trained model applied to a NIR test image (Figure 6b). Across channels 0-7, different spectral bands capture varying information about the same substances, with VIS channels showing consistent contrast while NIR channels 5-7 exhibit reduced contrast for certain materials. Both spectral ranges demonstrate effective anomaly detection through reconstruction error analysis, with quantitative performance comparisons presented in Table 1.

Looking at the reconstruction-based results shown in Figure 7, the autoencoder performance demonstrates successful capability across both datasets. Both ViT and ResNet autoencoders reconstruct normal road surfaces while producing high reconstruction errors (shown in darker regions) where substances are applied. The predicted anomaly maps closely match the ground truth, indicating effective thresholding and morphological post-processing.

For the MVTec Hazelnut dataset, the models demonstrate capability in detecting surface defects like cracks and holes, with reconstruction loss maps highlighting anomalous regions and binary predictions showing spatial correspondence to the ground truths. The self-supervised approaches shown in Figure 8. Barlow Twins produces more focused, localized high-anomaly regions around the defect, while SimCLR generates broader anomaly distributions. However, the heatmaps of both approaches are not centered around the defect, materials and include areas that are not anomalies.

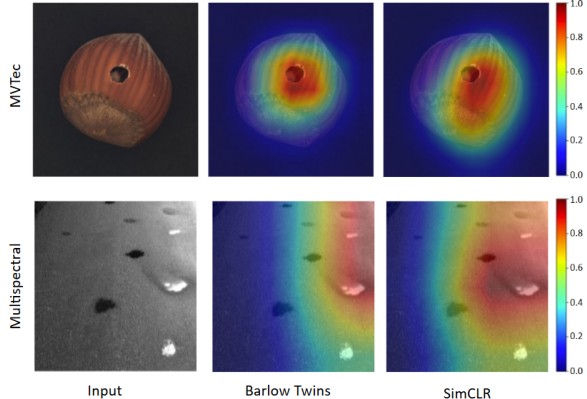

**Figure 8. Comparison of self-supervised anomaly detection methods**t. Both methods tend to localize the defect and materials through high anomaly scores.

## 4.5 Detection Performance

The performance analysis reveals several key findings. For the multispectral road dataset, the results are shown in Table 1. The ViT Autoencoder outperforms ResNet across all metrics, with VIS images (F1: 0.67) slightly outperforming NIR images (F1: 0.64). High recall values (0.86-0.87) indicate excellent anomaly detection sensitivity. For our application, high recall values are prioritized over precision, as they reflect the proportion of correctly detected anomalies — crucial since only identified anomalies can be further analyzed. The lower precision values

**Table 1. Results on Multispectral Road Dataset (VIS vs NIR Images)**

| Method | Precision | | Recall | | F1 | | IoU | | AUC-ROC | | AUC-PR | |
|---|---|---|---|---|---|---|---|---|---|---|---|---|
| | VIS | NIR | VIS | NIR | VIS | NIR | VIS | NIR | VIS | NIR | VIS | NIR |
| ResNet AE | 0.48 | 0.45 | 0.80 | 0.78 | 0.60 | 0.57 | 0.43 | 0.40 | 0.58 | 0.55 | 0.35 | 0.32 |
| ViT AE | **0.55** | **0.51** | **0.87** | **0.86** | **0.67** | **0.64** | **0.50** | **0.47** | 0.67 | 0.63 | **0.45** | **0.42** |
| SimCLR | - | - | 0.46 | - | 0.63 | - | 0.46 | - | **0.69** | - | - | - |
| Barlow Twins | - | - | 0.23 | - | 0.37 | - | 0.23 | - | 0.67 | - | - | - |

**Table 2. Results on MVTec RGB Hazelnut Dataset**

| Method | Precision | Recall | F1 | IoU | AUC-ROC | AUC-PR |
|---|---|---|---|---|---|---|
| ResNet AE | 0.65 | 0.63 | 0.64 | 0.51 | *0.85* | 0.47 |
| ViT AE | **0.75** | **0.65** | **0.70** | **0.57** | **0.88** | **0.54** |
| Barlow Twins | *0.67* | *0.64* | *0.65* | *0.52* | 0.82 | *0.49* |
| SimCLR | 0.54 | 0.60 | 0.57 | 0.43 | 0.78 | 0.42 |

(0.51-0.55) in combination with the IoU scores (0.47-0.50) show that most anomalies are found, but only partly detected. AUC-ROC scores (0.63-0.67) show reasonable discrimination capability, as indicated by Davis and Goadrich [31]. AUC-PR scores (0.42-0.45) show moderate performance in the anomaly detection task, which is expected given the rarity of anomalous pixels in road surface images. The self-supervised methods show contrasting performance: SimCLR achieves the highest AUC-ROC (0.69) with moderate recall (0.46) and F1 score (0.63), demonstrating better overall discrimination capability. In contrast, Barlow Twins shows significantly lower performance across all metrics (F1: 0.37, recall: 0.23), suggesting that redundancy reduction objectives may be less suitable for this anomaly detection task compared to contrastive learning approaches.

The MVTec Hazelnut dataset results are presented in Table 2. Notably, the ViT autoencoder achieves the highest performance across most metrics, with a precision of 0.75, an F1 score of 0.70, and an AUC-PR of 0.54, indicating its effectiveness in detecting anomalies. The ResNet and Barlow Twins also show competitive performance, with the ResNet achieving the second-highest AUC-ROC score of 0.85, suggesting that traditional autoencoder architectures can still be effective in certain scenarios. In contrast, SimCLR performs relatively poorly, suggesting that the chosen contrastive learning approach may not be well-suited for this specific task.

## 5 Conclusion and Future Work

In this paper, we presented an approach for material and anomaly detection using deep learning for autonomous driving and environmental monitoring applications. We implemented and evaluated our approach on multispectral and MVTec hazelnut datasets and provided comparisons between ResNet [5] and Vision Transformer (ViT) [6] en-coders for autoencoder architectures, as well as SimCLR [7] versus Barlow Twins [8] for self-supervised learning.

The experiments suggest that ViT demonstrates better anomaly detection performance compared to ResNet architectures across both datasets. Reconstruction-based approaches prove more effective than distance-based methods for RGB anomaly detection tasks, VIS spectrum images provide slightly better detection performance than NIR for road surface anomalies, and multispectral information enables comprehensive anomaly detection by leveraging spectral signatures invisible to single-band imaging. The findings suggest that transformer architectures, particularly when combined with reconstruction-based learning, show advantages for multispectral anomaly detection applications.

Despite encouraging results, there is still room for improvement. Future work will: (i) classify specific materials (e.g., fertilizer, soil, sand, ethanol) and defect types (e.g., cracks, holes); (ii) enhance reconstruction with combined losses, such as SSIM [33] plus MSE; (iii) assess alternative distance metrics, including k-NN–based scoring [34]; (iv) adopt augmentation-robust self-supervision—because SimCLR and Barlow Twins depend heavily on augmentations that are challenging for multispectral data, we will explore DINO [35] and MAE [36]; and (v) integrate additional spectral bands (e.g., SWIR) to improve material discrimination.

## Acknowledgments

This work has partially been funded by the Deutsche Forschungsgemeinschaft (DFG, German Research Foundation) under Germany's Excellence Strategy, EXC-2070 - 390732324 - PhenoRob. Further, it has been funded by the Deutsche Forschungsgemeinschaft (DFG, German Research Foundation) - RO 4839/6-1 - 459376902 - AID4Crops.

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

# A   Unsupervised Learning Details

**ResNet Encoder**.   The ResNet-based autoencoder follows a convolutional design inspired by [37], where the encoder transforms input images of size $C \times 128 \times 128$ into a latent space of size $512 \times 16 \times 16$. The encoder begins with a $7 \times 7$ convolution expanding channels from $C$ to 64, and applies a sequence of convolutional blocks with batch normalization and ReLU activation to progressively reduce spatial resolution while extracting hierarchical features.

**Vision Transformer Encoder**. The ViT-based autoencoder replaces the convolutional encoder with a transformer-based design. The ViT-B/16 model [6] divides the input into 64 non-overlapping $16 \times 16$ patches, each linearly embedded and augmented with a class token and positional encoding. The encoder comprises 12 transformer blocks with multihead self-attention (MHSA) and multilayer perceptrons (MLPs), enabling global context modeling [38] across the image.

Finally, the decoder progressively reduces the feature map's channel size ($256 \rightarrow 128 \rightarrow 64$) while spatial resolution is doubled at each stage ($16\times16 \rightarrow 32\times32 \rightarrow 64\times64 \rightarrow 128\times128$). The final layer-sigmoid- produces the final reconstruction map with dimensions $C \times 128 \times 128$.

# B   Self Supervision Learning Details

The shared encoder architecture follows the standard ResNet50 design, beginning with initial convolutional processing through convolutions, batch normalization, ReLU activation, and max pooling operations. This is followed by four sequential residual block layers that progressively extract hierarchical features at different abstraction levels. While both methods use the same encoder, they differ in their projection head designs. The SimCLR implementation features a streamlined two-layer projection head, while Barlow Twins uses a three-layer projection head that transforms representations into the projection space. (Figure B.1) shows the complete pipeline for anomaly detection using SSL approach, while in (Figure B.2) an illustration about the Mahalanobis distance.

# C   Training Settings and Hardware Configuration

All models select the best checkpoint based on validation loss. All experiments are conducted on the CubeSat computational cluster at the University of Bonn, equipped with four NVIDIA GeForce GTX 1080 Ti GPUs, each with 11GB VRAM, and 125GB RAM. The models are implemented using Python 3.8.10, PyTorch 2.4.1, and CUDA 11.6 for GPU acceleration.

Training configurations are optimized for each architecture to ensure fair comparison across methods. Autoencoder models are trained for 200 epochs with a batch size of 8, using the Adam optimizer and MSE loss. The ResNet autoencoder uses a learning rate of $5 \times 10^{-4}$ with weight decay of $1 \times 10^{-5}$, while the ViT autoencoder requires a lower learning rate of $5 \times 10^{-5}$ with higher weight decay of $1 \times 10^{-4}$ due to its transformer architecture.

For self-supervised methods, Barlow Twins trains for 150 epochs with batch size 8, learning rate $5 \times 10^{-4}$, embedding dimension 2048, lambda parameter 0.01, weight decay $1 \times 10^{-6}$, and gradient clipping at norm 1.0. SimCLR requires more extensive training with 500 epochs and a larger batch size of 32, using a learning rate $5 \times 10^{-4}$, a projection dimension of 128, a temperature 0.07 for the NTXent loss, weight decay $1 \times 10^{-4}$, and gradient clipping at norm 0.5. The longer training period and larger batch size for SimCLR are necessary due to the contrastive learning requirements. All models select the best checkpoint based on validation loss.

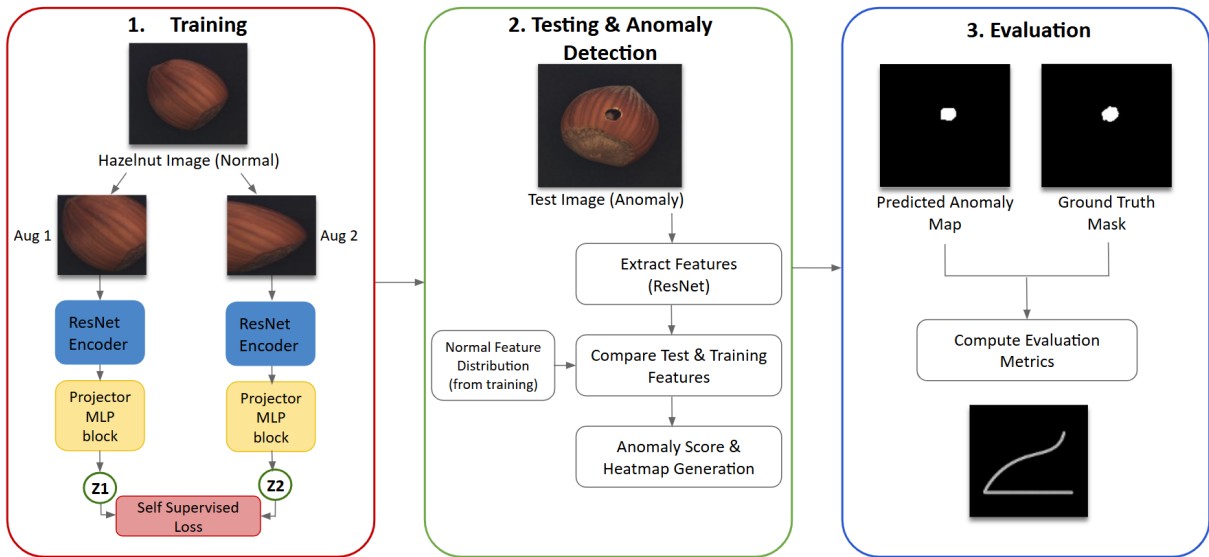

**Figure B.1. Self-supervised anomaly detection on RGB Hazelnut**. (1) Training: Augment normal images and learn representations with a ResNet-50 encoder and projection head; fit a Gaussian to the resulting normal features (assumed Gaussian-distributed). (2) Detection: Run test images through the trained encoder and compute Mahalanobis distances to the Gaussian to produce anomaly scores and heatmaps. (3) Evaluation: Compare predicted anomaly maps with ground-truth defect masks.

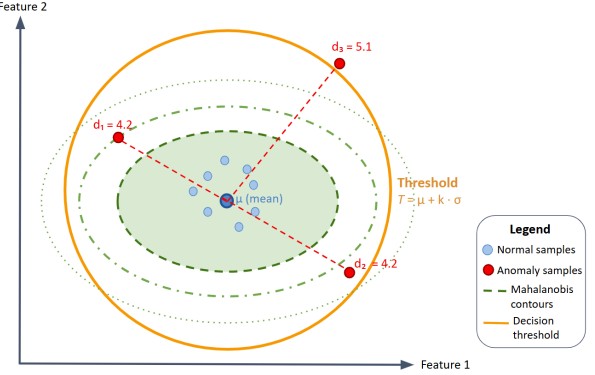

**Figure B.2. Mahalanobis distance-based anomaly detection in feature space**. Normal samples (blue) cluster around the distribution center $\boldsymbol{\mu}^*$, while anomalous samples (red) exhibit larger Mahalanobis distances. The decision threshold $T = \mu_v + k \cdot \sigma_v$ separates normal from anomalous regions, with contours representing equal Mahalanobis distance levels.

# D    Qualitative Analysis of Computational Complexity

While a full analysis is beyond the scope of this paper, we provide a qualitative comparison of the training and inference costs of autoencoders and self-supervised learning (SSL) methods. Training SSL models generally requires many more epochs to reach stable convergence. For example, SimCLR and Barlow Twins often need 500+ epochs, whereas reconstruction-based autoencoders converge much faster. SSL methods also demand significantly more RAM due to their reliance on large batch sizes (>256) and the use of dual encoders to generate different views, making training substantially more expensive.

During inference, SSL models are typically faster because the anomaly score is computed directly from precomputed feature statistics. In contrast, autoencoders require a full forward pass through both the encoder and decoder, which increases computational cost and latency.

