# OpenReview forum: "Self-Supervised and Unsupervised Multispectral Anomaly Detection for Unknown Substance and Surface Defect Identification"
_NLDL.org/2026/Conference — NLDL 2026 Poster_

### Official Review · Reviewer_8tkE · 2025-10-06
**Demonstrates a comparison of self-supervised and unsupervised anomaly-/surface defect-detection, with a comparison of performance of unsupervised methods on novel multispectral data**

**Rating:** 5
**Confidence:** 4
**Final Rating:** 5
**Final Confidence:** 4

**Summary:**

Describes methods for anomaly detection using self supervised learning (SSL), i.e. SimCLR and Barlow Twins, as well as
comparing these to supervised AutoEncoder (AE) methods, namely ResNet AE and ViT AE.
The paper aims to contribute the following:
- perform a first comparison of ViT-based and CNN-based autoencoders with contrastive learning for surface defect detection.
- show that ViT AE outperform ResNet and convolutional AEs on this task,
- demonstrate that Barlow Twins (BT) is superior to SimCLR for anomaly detection, particularly in computationally constrained settings.

The ViT is compared to ResNet on a novel dataset, containing multispectral images of spillage on a road.
This approach for creation and collection of a multispectral road (MSR) dataset is well justified with references
to earlier research showing that models with input from Near Infrared (NIR) can outperform those trained
solely on RGB.

Additional comparisons are performed using the MVTec AD Hazelnut (RGB) dataset, which contains augmented
images of hazelnuts with anomalies on the shell.
All of 4 models are compared on the MVTec AD dataset, while the 2 unsupervised methods (VIT,ResNet) are compared
on the novel MSR dataset.

Of the assessed models on the MVTec dataset, the lowest performer was the SimCLR, with the best-performing
being the ViT, and a slightly better performance for BT compared to ResNet AE.

Unfortunately there was no SSL performed on the MSR, with cited concerns that the augmentation
schemes that BT and SimCLR "heavily depend on" are challenging in the multispectral case.
This I consider to be the only weak claim in the paper, given that there
are still a significant amount of augmentations that would be compatible with BT / SimCLR
that remain available.
Arguably then the claim that the prohibited augmentations are the most significant for
BT/SimCLR needs citation, showing some benchmark that supports the claim.

The paper concludes that reconstruction-based methods seem more effective than distance-based methods
for the RGB anomaly detection tasks, and that models trained on VIS performs slightly better than
the ones trained with NIR.

**Strengths:**

Overall well written, easy to follow paper.

Makes a good case for prefering Barlow Twins over SimCLR w.r.t SSL anomaly detection, while alltogether
stating preference for reconstruction-based methods and ViT.

Seems to have a clear and consistent experimental setup, and well documented findings.

Conclusions seem well funded, and the future work seems interesting.

**Weaknesses:**

Since it does not seem like the self-supervised methods have been applied to the multispectral dataset, the title seems a bit misleading.
Similarly, this is unclear in figure 1, where it seems that SimCLR and BT are applied to the MSR dataset.

The claim that BT / SimCLR cannot be performed on the MSR dataset due to some incompatibility of the data augmentations
needs a supporting citation. (e.g. a benchmark like https://doi.org/10.14428/esann/2025.ES2025-142, where the authors can
show why a limited subset of augmentations like blurring, translation, or distortion is not possible to do)
I believe that a partial set of augmentations, with comments on which augmentations to avoid, may have provided
a little extra to the paper as a whole.

annoyance: figures should be [tb], i.e. top or bottom, rather than occuring in the middle of text blocks (e.g. figure 2, figure 3)
which creates strange virtual reading lines (it makes it seem like the text around may be divided into upper and lower block rather
than as a left/right column) - this was stated in the styleguide for the papers.

minor issue: acronyms (VIS,NIR) should be defined on first occurence, (e.g. suggest using latex acronyms and \gls{}, to help ensure these
are not left until section 3).

**Final Justification:**

After reading the other reviews and rebuttals, my rating remains positive.

Certain concerns regarding the readiness of the multispectral approach have been raised by other reviewers, but it seems clear from the paper that this is an early result on a relatively small (labeled) dataset.
Similarly, reviewers have raised questions as to the robustness of statistical conclusions or generalisations that one can get from a smaller dataset. I do not share those concerns at present moment, due in part to the experimental nature of the methods described, but also since I do not interpret the authors as making a claim towards generalisation across other forms of anomaly detection with multispectral data.

The authors also indicate that they will evaluate the self-supervised methods on the multispectral road data before a final revision, which I think will further strengthen the paper. Even better if this further includes an evaluation of the augmentation methods that may be suitable for the MSR data. This will make the comparison of methods easier, and the paper more coherent as a result.

In conclusion, I believe the contribution of the paper remains a clear one, and that any concerns or corrections that I could see to this paper will be improved in the revised version.


PS. Apologies for a forgotten concern during the initial review, but for the plots it could be a good idea to make use of a more universally readable colormap ref: https://www.nature.com/articles/s41467-020-19160-7 (alternative direct-link https://www.fabiocrameri.ch/colourmaps/)

**Justification:**

Overall the paper is interesting and provides a novel multispectral road (MSR) dataset.
It provides clear results towards their proposed contributions
- comparing two unsupervised methods (ResNet and Vision Transformer
based autoencoders) and two self-supervised (SSL) (Barlow Twins and SimCLR) on surface defect detection.
- showing that ViT outperforms the other methods on the task
- demonstrating that Barlow Twins outperforms SimCLR for anomaly localisation.

The weaknesses of the paper are minor, though particularly the dismissal of self-supervised learning (SSL) on
grounds of challenging augmentations on the MSR dataset was not entirely convincing.

---

> ### Author Rebuttal · Authors · 2025-10-22
>
> We sincerely appreciate your thoughtful feedback and the constructive insights provided for our manuscript. Your comments motivate us to continue refining and enhancing the manuscript to better contribute to this important field.
>
> 1. Evaluation on Multispectral dataset
>
> We agree with the reviewer. Upon acceptance of the paper, we will include additional results evaluating the performance of the self-supervised approaches on the multispectral data. For the figure, we will enhance it.
>
> 2. Augmentation of MSR data
>
> We will add this paper to our citations and provide more results once the paper is accepted on the MSR dataset. Furthermore, we will provide an overview why at specific domains not all of the augmentations used for the normal computer vision benchmarks work in those cases.
>
>
> 3. Figure editing and Minor issues
>
> We will enhance the structure of the paper according to this comment to make it much more presentable. Regarding the acronyms, we will define them as early as possible in the paper.

---

### Official Review · Reviewer_Y6bb · 2025-10-08
**Multispectral anomaly detection. Interesting application and dataset limited by small-scale evaluation**

**Rating:** 2
**Confidence:** 4

**Summary:**

This paper presents an investigation of self-supervised approaches for anomaly detection in multispectral images. To do this, four different models (convolutional and ViT based autoencoders, SimCLR, and Barlow Twins) are trained without anomaly annotations. The authors distinguish between the autoencoder models (which they call "unsupervised") and the other models (which they call self-supervised). SimCLR and Barlow Twins are used with a distance-based anomaly detection strategy, while the autoencoders are used with a reconstruction based (pixel level) anomaly detection strategy.

The paper also presents a new dataset collected using a sensor array system mounted on a car. It collects multispectral images with 9 bands, including RGB and NIR. The system collects images of road surfaces and for the training data, only clean road surfaces are used. For the evaluation data, different contaminations in the form of spills of different substances are put on the road surface.

The models are also evaluated on an existing dataset, the MVTec hazelnut dataset, which is RGB only.

The ViT outperforms the other methods in all metrics, and the reconstruction-based anomaly detection seems to outperform the distance-based. All approaches seem to have a similar performance with regards to recall, but they differ substantially in precision. The authors argue that recall is more important in this application, which would incur that the difference between the models is small.

**Strengths:**

The authors provide a clear and detailed methodology, covering data acquisition with a custom sensor array, pre-processing, and a well-structured comparison of four distinct deep learning models. This gives the paper clarity and potential for reproducibility.
The major novelty lies in the multispectral dataset.
The evaluation is performed on two distinct datasets.
The paper's primary strength is its focus on a practical and important real-world problem—road safety and environmental monitoring—which is often overlooked in theoretical ML research. The creation of a new, albeit small, multispectral dataset for this task is a valuable contribution.
The evaluation looks good. Uses reasonable evaluation metrics, and compares several methods.

**Weaknesses:**

The paper does not compare anomaly detection using multispectral data and RGB data. It would be easy to drop all channels except the RGB channels and compare the results.
A major weakness is that the SimCLR, Barlow Twins were only evaluated on the MVTec RGB dataset, not on the primary multispectral road dataset. The authors' claim that reconstruction-based methods are superior is therefore only validated for the RGB domain, and their performance on the main task remains unknown.
The paper does not quantify the computational requirements of the models.
I would say that autoencoders are also self-supervised. The fact that the authors make the distinction and puts the labels on the two classes of models gives a slightly confusing impression.
The test set it small; it contains only 18 images.
The findings are limited by the highly controlled nature of the dataset. The anomalies consist of neatly placed 'placeholder substances' on a clean road surface. The study does not evaluate the models' robustness to real-world complexities such as varying weather, lighting conditions (shadows, rain), irregular spill shapes, or more subtle anomalies like oil slicks, which limits the conclusions about the method's practical applicability.

**Justification:**

I have rated this paper as a '2: Reject'. The paper does introduce a novel and interesting application for anomaly detection with a new multispectral dataset, a clear contribution to the community. The experimental comparison between ViT and ResNet autoencoders is well-executed and provides valuable insights.

However, the paper's contributions must be viewed as preliminary due to several significant weaknesses that temper the results. The conclusions are drawn from a very small test set of only 18 images, which raises concerns about statistical robustness. Furthermore, the study critically lacks a direct baseline comparison against an RGB-only model on the multispectral data, making it difficult to quantify the true benefit of the extra spectral channels. Finally, the lack of any computational performance metrics makes it impossible to assess the system's viability for the stated real-time application.

While the paper is not robust enough for an unconditional accept, its novelty and interesting direction could merit publication if the weaknesses were addressed. I recommend rejection but I think it is close to accept.

---

> ### Author Rebuttal · Authors · 2025-10-22
>
> We sincerely appreciate your thoughtful feedback and the constructive insights provided for our manuscript. Your comments motivate us to continue refining and enhancing the manuscript to better contribute to this important field.
>
> 1. Evaluation on Multispectral dataset
>
> We agree with the reviewer. Upon acceptance of the paper, we will include additional results evaluating the performance of the self-supervised approaches on the multispectral data.
>
> 2. Superior performance of reconstruction-based methods
>
> We will augment our conclusion for the final version once the paper is accepted by new results as mentioned previously.
>
>
> 3.a. Computational cost / Categorization convention
>
> We will include an additional table summarizing the computational requirements of the different methods, including latency, memory footprint, and throughput. Our preliminary intuition suggests that autoencoders require less training time compared to self-supervised methods, while exhibiting a relatively higher memory footprint and inference time.
> Regarding the classification of autoencoders as self-supervised methods, we acknowledge this ambiguity in the literature—some works consider them a distinct category, whereas others group them under self-supervised learning. We address this point explicitly in the paper.
>
> 3.b. Data Availability
>
> The dataset was designed to be experimental. Larger-scale annotation remains an ongoing effort.
>
> 4. Real world conditions
>
> We agree with the reviewer’s observation. In future work, we plan to conduct more extensive experiments under various real-world conditions by performing covariate shift analyses.

---

### Official Review · Reviewer_X3hs · 2025-10-10
**Comparison of self-supervised and unsupervised methods for multispectral anomaly detection**

**Rating:** 4
**Confidence:** 4

**Summary:**

The paper presents a study of self-supervised and unsupervised DL methods for anomaly detection on multispectral imagery, focusing on identifying unknown materials and detecting surface defects. Four methods are evaluated: autoencoders with ResNet and Vision Transformer encoders, and two self-supervised approaches (SimCLR and Barlow Twins). The models are tested on both a proprietary multispectral road dataset and the public MVTec hazelnut dataset. The ViT-based autoencoder demonstrates superior anomaly detection performance over the ResNet-based and convolutional counterparts, while Barlow Twins outperforms SimCLR in terms of anomaly localization. The approach eliminates the need for labeled anomaly data and demonstrates strong recall, along with spatial correspondence with ground truth labels, highlighting its value for autonomous systems and environmental monitoring.

**Strengths:**

* Novel use case of ViT-based autoencoders for open-world unknown substance and defect identification on multispectral data.
* Experiments are conducted on two datasets—one real-world multispectral (with physical proxy contaminants) and the public MVTec Hazelnut dataset—demonstrating generalizability.
* ViT autoencoders achieve the highest detection scores (F1 up to 0.70 on RGB and 0.67 on multispectral data). Barlow Twins also shows more precise localization than SimCLR. High recall ensures most anomalies are flagged, which is critical for safety applications.
* The paper details data collection and augmentation, model architecture, and evaluation pipeline, along with relevant statistical and morphological postprocessing steps.

**Weaknesses:**

* The primary multispectral dataset has only 18 labeled test images, which may limit the robustness of statistical conclusions and generalization to other unknown substances or conditions.
* While recall is strong, precision and IoU for the best models remain moderate (precision ≤0.55, IoU ≤0.50 on multispectral), meaning some normal pixels may be misclassified.
* The impact of individual architectural choices (e.g., decoder, loss functions) is not deeply dissected, and only two self-supervised methods are tested without more recent approaches like DINO or MAE.
* SimCLR required much larger batch sizes and longer training times, suggesting some methods may not be practical for real-world deployments with limited computational capacity.

**Justification:**

The paper provides a novel and clearly written comparison of recent self-supervised and unsupervised methods for anomaly detection in a challenging open-world setting, using real multispectral imagery and public datasets. The reported improvements—especially from ViT-based autoencoders and Barlow Twins—are compelling and sufficiently validated with robust metrics, despite some dataset size limitations. The work meaningfully advances anomaly detection but can benefit from deeper ablations and expanded datasets to improve reproducibility and reliability in future iterations.

---

> ### Author Rebuttal · Authors · 2025-10-22
>
> We sincerely appreciate your thoughtful feedback and the constructive insights provided for our manuscript. Your comments motivate us to continue refining and enhancing the manuscript to better contribute to this important field.
>
> 1. Data Availability
>
> We acknowledge that the annotated subset of the multispectral dataset (18 test images) is relatively small. This limitation arises because the anomaly detection task is pixel-wise and the substances occupy only a small fraction of each image. High-quality annotation therefore requires sub-centimeter precision and manual verification across eight spectral bands, which is considerably more time-consuming than labelling at image level. Larger-scale annotation remains an ongoing effort.
>
> 2. Balance of Precision & Recall
>
> So far, our focus has been on minimizing missed anomalies, which led us to select a threshold that prioritizes high recall—even at the cost of lower precision. While alternative thresholds could offer a better balance between recall and precision, we determined that maximizing recall was more critical for our objectives.
>
>
> 3.a. Model Architecture
>
> We experimented with conventional ResNet-18 and ViT-B/16 encoders while keeping the decoder architecture identical for both. This setup aimed to provide initial insights into anomaly detection using different backbone types, without introducing additional variables that could influence performance
>
> 3.b. Future Methods
>
> We acknowledge this concern. In future work, we will integrate additional approaches as suggested by the reviewer. Notably, this study represents the first attempt to apply self-supervised learning methods to hazardous material detection framed as an anomaly detection problem.
>
> 4. Computational Considerations
>
> We agree with the reviewer’s observation. We recognize the need for sufficient data and computational resources to effectively train SimCLR-based models. In future work, we will explore more computationally efficient self-supervised learning approaches, such as:
>
> [1] Tong, Shengbang, Yubei Chen, Yi Ma, and Yann LeCun. “EMP-SSL: Towards Self-Supervised Learning in One Training Epoch.” arXiv preprint arXiv:2304.03977 (2023).

---

### Meta-Review · Area_Chair_hDkK · 2025-11-01

**Recommendation:** Accept (Poster)
**Confidence:** 4

**Metareview:**

Despite some weaknesses - including the limited dataset size, lack of experiments/validation - the paper’s strengths, such as its novelty and clear, detailed methodology, appear to outweigh these limitations by a **very small margin**. Overall, I recommend **borderline** acceptance of this paper.

---

### Decision · Program_Chairs · 2025-11-05

**Decision:**

Accept (Poster)

**Comment:**

We recommend a poster presentation given the AC and reviewers recommendations.